

# Coral disease prevalence estimation and sampling design

Eric Jordán-Dahlgren[1], Adán G. Jordán-Garza[2] and
Rosa E. Rodríguez-Martínez[1]

[1] Instituto de Ciencias del Mar y Limnología, Universidad Nacional Autónoma de México, Puerto Morelos, Quintana Roo, México
[2] Facultad de Ciencias Biológicas y Agropecuarias, Universidad Veracruzana, Tuxpan, Veracruz, México

Corresponding author
Eric Jordán-Dahlgren,
jordan@cmarl.unam.mx

## ABSTRACT

In the last decades diseases have changed coral communities' structure and function in reefs worldwide. Studies conducted to evaluate the effect of diseases on corals frequently use modified adaptations of sampling designs that were developed to study ecological aspects of coral reefs. Here we evaluate how efficient these sampling protocols are by generating virtual data for a coral population parameterized with mean coral density and disease prevalence estimates from the Caribbean scleractinian *Orbicella faveolata* at the Mexican Caribbean. Six scenarios were tested consisting of three patterns of coral colony distribution (random, randomly clustered and randomly over-dispersed) and two disease transmission modes (random and contagious). The virtual populations were sampled with the commonly used method of belt-transects with variable sample-unit sizes (10 × 1, 10 × 2, 25 × 2, 50 × 2 m). Results showed that the probability of obtaining a mean coral disease prevalence estimate of ±5% of the true prevalence value was low (range: 11–48%) and that two-sample comparisons achieved rather low power, unless very large effect sizes existed. Such results imply low statistical confidence to assess differences or changes in coral disease prevalence. The main problem identified was insufficient sample size because local mean colony size, density and spatial distribution of targeted coral species was not taken into consideration to properly adjust the sampling protocols.

# INTRODUCTION

Unprecedented coral disease epizootics have changed the structure and function of coral reefs (*Aronson & Precht, 2001*; *Bruno et al., 2007*; *Harvell et al., 2007*). The causes of these epizootics are linked to environmental degradation due to global warming (*Hoegh-Guldberg et al., 2007*; *Bruno et al., 2007*; *Eakin et al., 2010*), and local impacts, such as eutrophication (*Bruno et al., 2003*; *Kaczmarsky, Draud & Williams, 2005*), pollution (*Klaus et al., 2007*), overfishing (*Mumby et al., 2006*), sedimentation (*Rogers, 1990*; *Fabricius, 2005*), dredging and coastline modification (*Miller et al., 2016*), plastic waste (*Lamb et al., 2018*) and diving activities (*Lamb et al., 2014*).

Our ability to fully understand the dynamics of coral diseases on coral reefs is hampered by the complex functioning of the coral holobiont and the limited knowledge of coral-disease etiologies (*Ainsworth et al., 2011*; *Work & Meteyer, 2014*). Nevertheless, recent reports indicate that moderating or eliminating local impacts can reduce coral disease susceptibility (*Fabricius, 2005*; *Kaczmarsky, Draud & Williams, 2005*; *Lamb et al., 2014*). Therefore, sound assessments of the diseases effect are necessary to support expensive managing actions.

One of the most convenient disease assessments is to estimate prevalence (the proportion of affected individuals) and although simple in principle, prevalence estimates with acceptable accuracy and precision are not easy to obtain. Logistical constraints of underwater work make surveys on coral reefs a challenge and perhaps that is why proven and practical coral ecological-sampling protocols are often adapted to estimate disease prevalence (Table S1). However, by doing so, the sampling protocol is not specifically tailored to answer the question posed by a prevalence analysis which is related to a binomial condition (diseased or not). Two relevant statistically concerns arise: (1) the coral colonies sampled are not independent and (2) the necessary sample size is not properly estimated. These issues may result in low statistical power, implying the possibility of (1) low probability of accurate mean estimates and of finding true effects, (2) low reliability when claiming a significant effect and (3) an exaggerated estimate of the magnitude of the effect when this one is true (*Conner, McCarty & Miller, 2000*; *Salman, 2003*; *Nusser et al., 2008*; *Button et al., 2013*). To evaluate the importance of these issues when estimating coral disease prevalence in Caribbean key reef-building species, whose colonies tend to be large, we used a simulation approach parametrized by using real data on yellow-band syndrome affecting the coral *Orbicella faveolata* in a Mexican Caribbean reef. The sampling of virtual coral populations, with known disease prevalence, allowed us to evaluate the accuracy and precision of prevalence estimates obtained by commonly used sampling protocols, as well as their power in comparisons for a given effect size.

## METHODS

To parameterize the simulations we used mean density and prevalence estimates as obtained from ecological surveys adapted to include coral diseases and other signs of colony conditions. We selected the data from the reef building coral *O. faveolata* affected by the yellow-band syndrome in Mahahual reef, in the Mexican Caribbean, in July 2006. The sampling protocol for the site consisted of four sampling stations in the fore-reef, separated from each other by at least 500 m, in a depth range of 10–15 m. Each station comprised six 25 × 2 m belt-transects that were randomly deployed in an area of aproximately 100 × 100 m. The mean prevalence of the yellow-band syndrome in 756 *O. faveolata* colonies surveyed in 24 belt-transects (1,200 m$^2$) at the Mahahual reef site was 16.2%.

### Simulated scenarios

Point pattern distributions (R spatstat, *Baddeley, Rubak & Turner, 2015*) were generated in a sampling frame area of 1,500 m by 200 m, by using a point density of 0.63 colonies m$^{-2}$

and a proportion of "diseased" points of 0.162. Each point represented the center of a simulated coral colony, thus equivalent to real surveys where only colonies whose centers lie within the boundaries of the sample unit (i.e., belt-transect) are included (*Zvuloni et al., 2008*). Given that the spatial distribution of *O. faveolata* at the site is unknown, six different scenarios were simulated, based on three spatial point pattern distributions (random, randomly clustered, and randomly over-dispersed), each of them with two disease transmission modes (random and contagious). The random-disease transmission mode simulation pertains to water-borne transmission with no particular resistance on the host population and every colony had the same probability of being diseased (*Jolles et al., 2002*). The simulated diseased points were marked by means of a random binomial generator (package binom in R, *Dorai-Raj, 2014*). To simulate a possible contagious situation the probability of being diseased was dependent on the distance to a diseased neighbor, thus a nearest neighbor distance criteria determined the probability of transmission. Given the lack of better information, we subjectively assigned the probability of being diseased as a function of decreasing nearest neighbor distance percentiles (0.5, 0.7, 0.8, 0.9 and 0.99) and used these probabilities on a random binomial generator to mark the diseased colonies on the virtual populations. In this way the closer the colonies were to each other, the higher the probability of being assigned a disease mark. Given the random processes involved on these simulations, the final coral density and disease prevalence varied around our initial target values, so to verify that the expected types of distribution pattern and disease transmission modes were achieved, we employed plots (Figs. S1–S3) of Monte Carlo confidence bands for Bezag's transformation of Ripley's K (*Baddeley, Rubak & Turner, 2015*; see Supplementary Material codes: 1. Scenarios & Ripley, R). Ripley's K is a spatial analysis technique that estimates the expected number of points within an increasing radius *r* from an arbitrary point and Besag's L is the normalization of K by way of square root transformation. With this technique the point distribution patterns (random, clustering or overdispersion) can be described as a function of distances by comparing deviations to a completely spatial randomness condition whose value after normalization is 0. Thus positive values indicate clustering while negative ones overdispersion.

## Choosing a single scenario to perform simulated surveys

Given that we had no information on the spatial arrangement neither of *O. faveolata* in general or of the diseased colonies in particular, we tested the six simulated scenarios with a similar sampling protocol as the one used in situ (i.e., 24 transects of 25 × 2 m in four stations) to determine which of the six scenarios better reproduced mean density and prevalence field estimates (see Supplementary Material codes: 2. Sampling & estimates) Although our own estimates might be inaccurate, we took this approach to be able to simulate the case of a single coral host that has relatively low densities, but can reach large sizes, and is affected by a particular disease sign. A total of 30 independent surveys per scenario were performed to obtain colony density and prevalence estimates, each survey estimates were compared to the real field estimates. The abundance comparisons were done with negative binomial generalized linear models (R package MASS,
*Venables & Ripley, 2002*), that were validated if no significant residual overdispersion was present, estimated as 1-pchisq (model null deviance, model residual degrees of freedom). For disease prevalence comparisons, quasibinomial Generalized Linear Models (GLM's) were used and considered acceptable if there was no significant over dispersion of standardized deviance residuals within 95% confidence envelopes (R package binomTools; *Christensen & Hansen, 2011*; see Supplementary Material codes: 3. Comparison of scenarios to Mh data.R). The scenario showing fewer significant differences in 30 trials, was selected for the performance tests of sampling protocols.

## Testing the accuracy and precision of common survey methods

To test the theoretical performance of coral disease prevalence sampling protocols, we compiled a list of methods reported in the literature (Table S1). Although sampling methodologies varied among the studies that were reviewed, the majority used belt-transects, as the sampling unit, and in average five belt-transects within a single sampling site. The size of the belt-transects in these studies was variable, but the most common dimensions were $10 \times 1$, $10 \times 2$, $25 \times 2$ and $50 \times 2$ m. To determine the precision and accuracy of coral disease prevalence estimates obtained by a protocol of five transects for each of the different belt-transect sizes, 100 independent samplings were carried out for each belt-transect size protocol (see Supplementary Material codes: 4. Accuracy of Estimates.R). Mean prevalence estimates were pooled to increase sample size and then calculate the credible interval for binomial means (R package binom; *Dorai-Raj, 2014*).

## Testing the achieved power of two sample comparisons

For each sampling protocol, we randomly select two out of the 100 surveys to estimate the achieved power in a two-sample comparison. Power was estimated by a power sample size function for proportions (R package stats, *R Core Team, 2016*). By repeating this procedure 100 times, predicted power curves as a function of effect size were generated as smooth splines, using generalized additive models adjusted for a beta regression distribution. Model validation was done through the package's gam check (R package mgcv; *Wood, 2011*). Effect size was determined by means of Cliff's delta, a non-parametric procedure that computes ordinal variables and thus is appropriate to deal with proportions (R package epiR; *Cliff, 1993*; *Stevenson et al., 2017* (see Supplementary Material codes: 5. Power and effsize in two sample comparisons.R). All statistical tests and simulation procedures were performed in R and RStudio. The code for the sampling simulation and accuracy estimates can be found in the Supplementary Materials.

# RESULTS

## Choosing a single scenario to perform simulated surveys observations

The reef scenario with a random coral-colony distribution and a random spatial pattern of diseased corals (R & rt) yielded similar results to the filed data estimates when sampled with 24 belt-transects of $25 \times 2$ m (Table 1). Similar results were obtained

**Table 1 Comparisons between field data and simulations of coral density and disease prevalence.**

| Scenario | Coral (point) mean density | Coral (point) mean disease prevalence |
|---|---|---|
| R & rt | 0.0 | 0.0 |
| R & ct | 26.7 | 0.0 |
| C & rt | 20.0 | 0.0 |
| C & ct | 36.7 | 16.7 |
| D & rt | 0.03 | 0.0 |
| D & ct | 16.7 | 0.0 |

for the over-dispersed coral colony distribution and the random coral disease transmission scenario (D & rt) (Table 1). Additional comparisons between these two scenarios showed redundant results and thus the R & rt scenario was selected to test the performance of the sampling protocols commonly used in coral disease prevalence surveys.

## Accuracy and precision of common survey methods

The accuracy and precision of the most common sampling protocols used in coral disease prevalence surveys (Table S1) was evaluated using five belt-transects of variable size ($10 \times 1$, $10 \times 2$, $25 \times 2$ and $50 \times 2$ m). For each one of these protocols the belt-transects were randomly deployed on a "site" of $100 \times 100$ m which was randomly selected within the $1,500 \times 200$ m sample frame of the R & rt scenario (mimicking the close layout of transects within a site). For each of the four protocols, 100 independent samplings were made. The precision and accuracy of the estimates varied widely within and between the different belt-transect sizes (Fig. 1). For the smallest sample unit design ($10 \times 1$ m), the estimated mean coral disease prevalence ranged from 0.1% to 36% and had large credible confidence intervals in all cases (Fig. 1). In this protocol the probability of obtaining a mean prevalence estimate of ±5% of the true disease prevalence value (16.2%) was 11%. In contrast, the mean disease prevalence estimated with the largest sample unit ($50 \times 2$ m) ranged between 10% and 21%, and had shorter credible intervals (Fig. 1). However, the probability of obtaining a mean coral disease prevalence estimate of ±5% of the true prevalence was still low (48%).

While for some surveys estimating gross disease prevalence is all that is required, the low accuracy and precision of the estimates imply that comparison between sites or times may not be reliable unless the expected effect size is large. To assess this possibility, we estimated the power for 100 independent two-sample tests for each protocol by randomly selecting pairs of their corresponding 100 samplings (Fig. 2). Calculated Cliff's delta effect sizes ranged from zero (no effect) to 0.8 (highest effect) and because tests are of similar context it is linearly related. We found that in all cases the power was positively related to the magnitude of the effect size and that, for a given effect size, the power of the test increased from the smallest to the largest sample unit as sample size increased (Fig. 2). Only two out of 100 comparisons of the largest sampling unit ($50 \times 2$ m) protocol reached the customary power level of 0.8. Furthermore, few tests had power values above 0.5 (less than 20%), whereas the power level of the large majority was

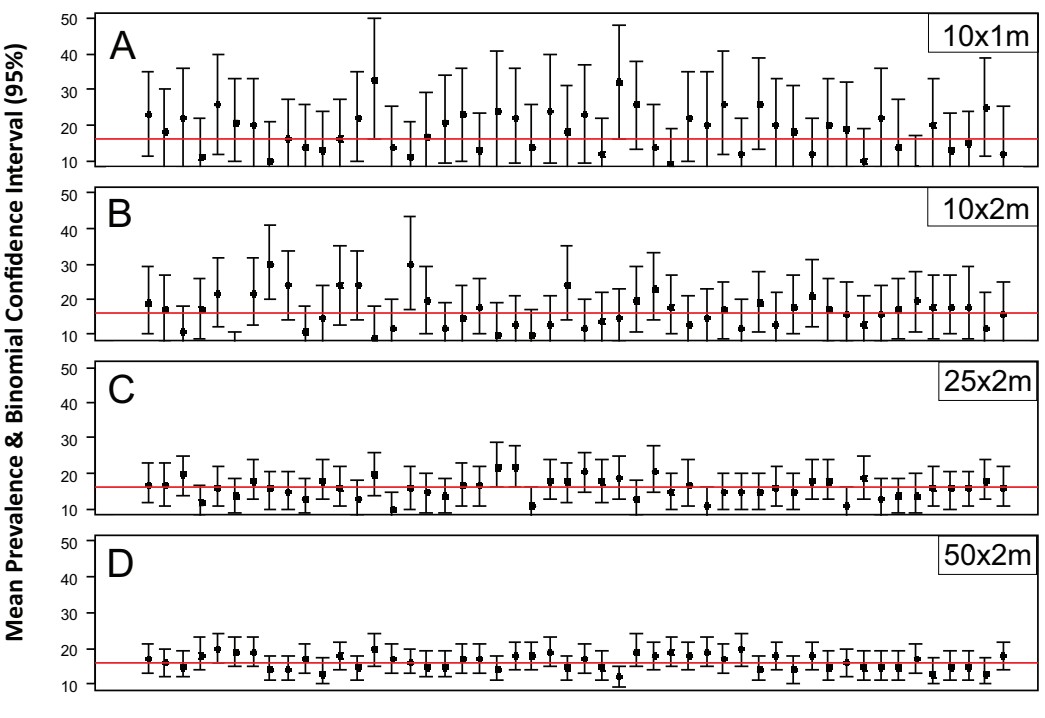

**Figure 1 Mean estimated disease prevalence.** Mean Estimated Prevalence (dots) and 95% Wilson binomial confidence interval (vertical bars) of 50 randomly selected surveys of the simulated 1,500 by 200 m reef zone. Each survey comprises one randomly deployed sample station/site (100 by 100 m), within the reef zone. Each sampling station consisted of five randomly deployed belt-transects of five dimensions: (A) 10 × 1m, (B) 10 × 2m, (C) 25 × 2m, (D) 50 × 2m.

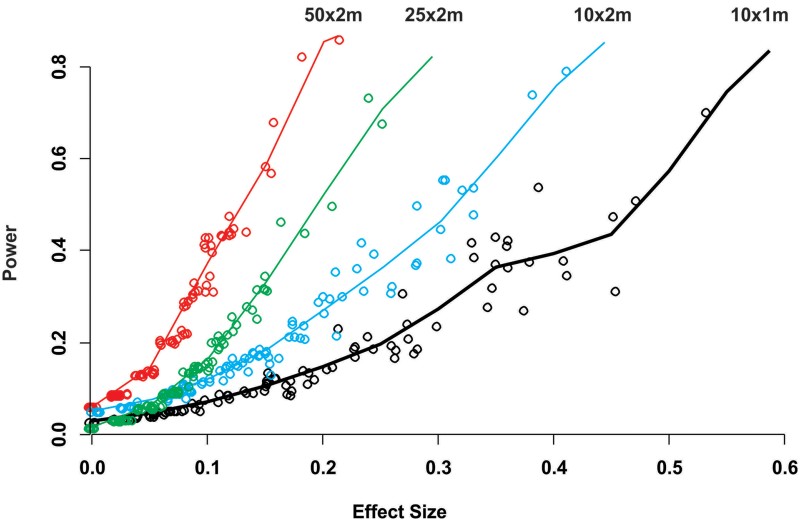

**Figure 2 Power of a two-sample comparison as function of the effect size for 100 two-sample station comparisons per sampling designs.** Open circles indicate the individual two sample power estimates. The solid line corresponds to a predict curve generated with a gam modeled with a beta regression distribution. Each sampling station consisted of five randomly deployed belt-transects and the upper text indicate the belt-transect dimensions for each curve and corresponding dat points set.

below 0.4 (Fig. 2). The smoothed predicted lines in the same figure shows that the low power trend is consistent for each belt-transect size.

## DISCUSSION

Coral reef surveys are usually conducted by SCUBA diving making it difficult to efficiently sample coral communities composed of multispecific assemblages living in a complex tridimensional structure. This problem has promoted the use of simple methods to conduct quantitative surveys of corals, such as the belt-transect and other type of quadrats as sample units. These methods have frequently been modified to assess coral disease prevalence (Table S1). The results of this work, however, suggest that this practice leads to poor accuracy with wide credible intervals when estimating coral disease prevalence of a host that can reach large sizes and thus relative low number of colonies per belt-transect area; a common case in Caribbean massive reef-building corals. In this situation, the power in two sample comparisons is low, unless very large effect sizes exist; thus hampering our ability to confidently assess changes in disease prevalence. As the power of a test is the likelihood that an effect will be detected, when indeed it exists, a high probability for a false negative (accepting the null hypothesis when it's false) is of much concern when conducting coral disease assessments.

We found that ecological survey methods that have been repeatedly tested to be reliable to assess community composition or benthic coverage (*Kinzie & Snider, 1978*; *Jokiel et al., 2015*) may be poor to estimate disease prevalence. The difference in efficiency lies in the purpose of the sampling design, ecological surveys are designed to address a multivariate problem, mainly the relative abundance of species or categories, which can't be reasonable sampled without the use of some sort of area encompassing method such as quadrats, transects, etc., that become the effective sampling unit. In contrast, disease surveys should be designed to deal with a single binomial univariate variable (diseased or not) most commonly of a single kind (species, forms, etc.) where the individual is the natural sampling unit, in our case a coral colony, and not the belt-transect. Therefore, unlike ecological surveys, in disease surveys the necessary sample size for a given precision and accuracy is a function of the proportion of diseased colonies and not of the number of belt-transect (or quadrats, circles, etc.). So, when using belt-transects they should be large enough to sample a sufficient number of colonies and, for this to occur, the size and shape of the belt-transects should be determined as a function of coral colony size and density of the target species within the sampled area. For instance, in the Mahahual reef survey, the mean transect abundance of *Agaricia agaricites* in our 25 × 2 m belt-transects was 243 colonies, but the mean abundance for the much larger *O. faveolata* colonies was of only 34 colonies. Estimation of disease prevalence in key reef building species that tend to have large colonies (i.e., *Acropora*, *Orbicella* and *Pseudodiploria* in the Caribbean) using protocols not specifically tailored for the properties of the target species, will likely result in too gross and inaccurate estimates; more so as the larger the colony, the fewer will be surveyed in a belt-transect.

While a sufficient sample size is indispensable to obtain a coral disease prevalence estimator, with a desired accuracy, the reliability of the estimate depends on obtaining representative samples (*Green, 1979*). In statistical sense representative samples can only be obtained if all coral colonies of the target species, within the sampling frame, have the same probability of being sampled (*Casella, 2008*; *Lohr, 2009*). However, when data are obtained by belt-transects, no other colonies outside the transects area are to be sampled and so colonies within the transect are not independently selected. Therefore, the magnitude of sampling bias could not be assessed and there is no way to estimate to what extent collected data represents the population under study, or to estimate the reliability of the prevalence values estimates (*Lohr, 2009*). Additionally, belt-transect deployment is usually done by convenience or selective criteria and while recognizing that very experienced researches may make a "more" representative deployment, it still is a non-probabilistic sampling approach that further increases bias and chance of pseudoreplication (*Hurlbert, 1984*; but see *Davies & Gray, 2015*). To our knowledge, only a few coral studies have made a probabilistic approach for coral sampling design (*Smith et al., 2011*), and thus statistically sound to generalize their results to the reef system under study.

A lack of representative samples or insufficient sample size may render impossible to detect valid statistical differences in coral disease prevalence between sites and through time, due to the low power of the tests. Moreover, low statistical power may also result in a deflation or inflation of the true effect size that result from sampling variation and random errors leading, in the case of inflation, to false positives when based on thresholds of statistical significance (*Button et al., 2013*). In this context, of hypothesis testing criteria, it should be remembered that rejection levels at $\alpha = 0.95$ and power $= 0.80$ have no other reason than custom. A power of 0.8 means that we are willing to accept a 20% chance of committing a type II error (a false negative), a margin of error that may be wide for critical coral disease decisions. For instance, if the interest is in detecting the effect of actions taken to minimize coral stress by reducing the residual waters output to the reef environment, a false negative may imply that no actions are taken or that planned actions are halted.

The results of this work allow us to suggest that coral disease prevalence surveys in the case of a single target species with low abundance should consist of several randomly sampling units deployed throughout the entire selected sample space, in order to ascertain representativeness and better estimate disease variability at the study site. If comparisons are to be made using belt-transects, the sample size design effect compensation due to non-random sampling should be determined, as sample size formulas assume simple random sampling. These recommendations will improve the quality of the estimates of coral disease prevalence, but will not allow making valid generalizations if quadrat-like sampling units are employed, unless sample-size and representativeness are statistically adequate. However, by minimizing the sources of bias stronger inferences could be made by issuing precise and credible statements (*Qian, 2014*), supported by detailed description of the sampling design rationale in relation to targeted species size and abundance within the selected sample space.

## CONCLUSION

Ecological survey methods commonly used to assess scleractinian coral community structure may be poor to estimate disease prevalence whenever large coral colony species are targeted (or rare species), unless the sample units used frequently (i.e., belt-transects, quadrats) are large enough. Insufficient sample size results in low power tests and statistical confidence when making comparisons in time and space. Coral disease prevalence surveys should be specific for a targeted species within a relatively homogeneous reef sample space in terms of its distribution pattern and colony size.

## ACKNOWLEDGEMENTS

We thank M.A. Maldonado, L. Vázquez-Vera, D. Baker and J. Andras for assistance in the field. Thanks to CRAN Foundation.

### Funding

This work was supported by the Coral Disease Working Group of the Global Environment Facility Coral Targeted Research program and by the Instituto de Ciencias del Mar y Limnología, UNAM. The funders had no role in study design, data collection and analysis, decision to publish, or preparation of the manuscript.

### Grant Disclosures

The following grant information was disclosed by the authors:
Coral Disease Working Group of the Global Environment Facility Coral Targeted Research program.
Instituto de Ciencias del Mar y Limnología, UNAM.

### Competing Interests

The authors declare that they have no competing interests.

### Author Contributions

- Eric Jordán-Dahlgren conceived and designed the experiments, performed the experiments, analyzed the data, contributed reagents/materials/analysis tools, prepared figures and/or tables, authored or reviewed drafts of the paper, approved the final draft.
- Adán G. Jordán-Garza contributed reagents/materials/analysis tools, authored or reviewed drafts of the paper, approved the final draft.
- Rosa E. Rodríguez-Martínez contributed reagents/materials/analysis tools, prepared figures and/or tables, authored or reviewed drafts of the paper, approved the final draft.

### Field Study Permissions

The following information was supplied relating to field study approvals (i.e., approving body and any reference numbers):
   Field survey was approved by SAGARPA (Agriculture, Natural Resources and Fisheries Secretariat).

## Data Availability

The raw measurements are provided in Supplementary File 1. The raw data shows the number of *Orbicella faveolata* colonies and those affected by the yellow-band syndrome recorded in 24 belt-transects (25 × 2 m) in Mahahual reef in 2006.

R codes used to create scenarios, sampling and estimates, comparison of scenarios, accuracy of estimates and power and effect size are provided.

## Supplemental Information

Supplemental information for this article can be found online at http://dx.doi.org/10.7717/peerj.6006#supplemental-information.

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
