# Peer review of "Coral disease prevalence estimation and sampling design"

_PeerJ, doi:10.7717/peerj.6006_

## Round 0.1 · original submission · Major Revisions

As you can see below, both referees are quite enthusiastic about your study and the contribution to the field. However, each also have a substantial number of comments and recommendations for improvement of the manuscript throughout. The most substantial of the comments is that the conclusions overstep the support of the data, which should be toned down as outlined by the second referee. PeerJ does not evaluate the novelty or importance of submissions, so there is no need to over-emphasize the conclusions. In my opinion, this paper makes an important contribution to the field, but as the referee points out, making claims beyond the scope of the results tends to take away from the paper. I expect that you should be able to address the concerns of the referees by revision to the manuscript, so I look forward to your resubmission.

·

Basic reporting

The study ‘Coral disease prevalence estimation and sampling design’ simulates belt-transect surveys of varying sizes on a simulated coral reef to determine how well current survey methodologies are able to accurately estimate disease prevalence. Their results indicate that it is difficult to accurately estimate disease prevalence with high confidence even when surveying large reef areas using the belt-transect method, which is currently the most commonly used method in field studies for studying coral disease. This study highlights an important issue in coral disease field studies and suggests that a probabilistic approach would allow for more accurate estimates of population level disease estimates. The authors provide sufficient background and context, reference the literature appropriately, and provide results that are relevant to their hypotheses. Overall, I recommend moving some figures and tables that are currently in the main text to supplemental information, and potentially adding a new figure if possible (see comments below). The writing includes several spelling mistakes and some grammatical errors, several of which I indicate in the comments below. I recommend considerable editing for clarity in the introduction, methods, and results sections, which would greatly improve readability of the manuscript and better highlight the value of this study. I recommend the authors re-submit this article with revisions according to the comments below.

Experimental design

The experimental design of this study is appropriate and the authors provide code and data to reproduce their results. The authors should reference supplementary information in the main text so readers know where to find associated code and results for each section of the study. The one issue I think the authors should consider in terms of their experimental design is that they created several scenarios (simulated coral reef landscapes) with specified coral densities, disease prevalence levels, and disease colony aggregations, yet selected a single scenario for the rest of this study based on the scenario that mostly closely estimated coral densities and disease prevalence in their field belt-transect surveys, yet their results suggest belt-transect surveys are unable to accurately estimate these measurements (see comment on lines 111-122 for further details). I do not think the study needs to be re-designed, but the motivation for selecting this experimental design should be more carefully considered and explained.

Validity of the findings

This is a novel study that highlights an important issue that is often ignored in coral disease field studies and provides well supported and validated results. This study was performed using simulated data and statistical analyses, which are made available in the supplementary information. The conclusions are well stated and link to the original research question.

Additional comments

Introduction
Line 37: Emergent diseases on coral reefs indicates that diseases only started arising since the 1970s on coral reefs and coral associated organisms. This sentence should be restructured to clarify that observations have only been reported since the 1970s although diseases were likely always part of the system.

Lines 37-62: This first paragraph can be written more concisely, there is not much information that is critical to the paper.

Line 53: the evidence of an association between coral disease and ocean acidification is spurious, consider removing.

Line 67: “logistic constrains” should be “logistical constraints”.
General: Species names should be italicized

Methods
Lines 82-89: Please edit this paragraph for clarity so that it is completely clear that the authors conducted 24 belt-transect surveys (6 surveys per sampling station, 4 sampling stations) and that abundance and prevalence estimates from those surveys were used to parameterize simulations.

Lines 82-83: change “parametrize” to “parameterize”, “estimates” to “to estimate”, and “reef build coral” to “reef-building coral”.

Line 84: Include the months when the surveys were conducted in 2006.

Line 87: Be consistent with writing the area of belt-transects: “25 by 2 m” versus “25x2 m”.

Lines 99-100: While Jolles 2002 argues that randomly aggregated diseased colonies could be associated with water-borne disease transmission, water-borne disease transmission is also hypothesized to be highly directional due to currents, tides, stream flow etc. I would recommend including additional conditions which would result in a random aggregation of diseased colonies, such as transmission by a vector that has a home range equal to or larger than the survey scale.
Lines 103-106: Does this approach mean that each diseased colony in the scenarios with contagious disease transmission was assigned a different probability of having a diseased nearest neighbor, or that there were multiple simulation scenarios and within each scenario diseased colonies had a single probability of having a diseased nearest neighbor? This information needs to be more clearly explained here. In addition, if the former approach was used (each colony was randomly assigned a different probability) I’m not sure this makes sense biologically because it would indicate that a single disease has five different transmission rates and that those differences in transmissibility are random across colonies. While it is great that the authors are considering that coral disease transmissibly given close distances among colonies is unknown within a range of possibilities, assuming that transmissibility varies by colony for a single disease is likely to produce less realistic scenarios rather than more realistic scenarios. Thus, if this approach was used I would suggest the authors re-run the simulations selecting a single nearest neighbor distance percentile or for each nearest neighbor distance percentile individually.

Lines 106-108: 1) Add 1-2 sentences to explain how Riply’s K estimates clustering and how the Bezag’s (Besag’s?) transformation of Ripley’s K accounts for greater variability associated with larger areas. 2) “Ripleys’ K” should be “Ripley’s K”. 3) Were Bezag’s transformation of Riply’s K plots created for all six scenarios? Based on this sentence it seems like they were, yet Figure 1 only shows results for one scenario (random clustering of colonies and random distribution of diseased colonies). Figure 1 should include these plots for all six scenarios. Figure 1 should also be moved to supplemental information as the information is important to ensure the simulated scenarios were created correctly, but do not provide any important information for the central question in the study.

Figure 1: The figure 1 title and caption need more information so that readers who are not familiar with Ripley’s K can easily and quickly interpret the results. Information should be added to briefly indicate the definitions of R, r, and L, what a departure of the black line from the shading would represent (clustering versus inhibition). In addition, I believe the y-axis represents L(r)-r rather than L(r) which cannot be discriminated with the current y-axis label, please add this information to the y-axis label.

Lines 111-122: This paragraph should be edited to make it clear that the goal of this comparison was to identify which colony distribution and disease transmission scenario most closely represented the coral reef where in situ estimates of colony density and disease prevalence were observed in order to identify one scenario to perform multiple simulated surveys for comparison in the rest of the study. I think this approach raises an important point for consideration. The authors have made several scenarios which were created based on specific criteria for colony density and diseased colony presence/aggregation (i.e., simulated coral reefs with known colony densities and disease prevalence), yet then selected the simulation that most closely resembles the coral reef density and disease prevalence estimated from in situ surveys, which they are arguing such methodologies may not be able to adequately capture estimates of coral density and disease prevalence. It seems like a very circular argument to me and could make the readers question the value of the study. Careful thought in how to present this approach could considerably improve the study motivation, approach, and value of results.

Lines 114-116: What does the scenario with fewer trials mean? How many trials? The first sentence seems to indicate that in each scenario 30 simulated surveys were conducted.

Lines 116-122: Explain model formulation more clearly here. Were the covariates simply the abundance or prevalence values observed in the belt transect surveys? Were the random effects field site or simulation scenario? It is not entirely clear here how this data was analyzed.

Table 1: This table should be moved to supplementary information. While it provides the motivation for the use of simulating surveys using the belt-transect method and what area should be covered in those simulated surveys, the information is not pertinent to the major results of this study.

Lines 138-139: Is there a reason for selecting two of every 100 surveys to assess power or was this number chosen arbitrarily?

Lines 145-147: Permit information could be probability be moved to “additional information and declarations”.

Results
Line 164: Change “100x100m” to “100 m x 100 m” or “100 m2”.

Figure 3: change “dat points” to “data points”.

Lines 185-186: I think this finding (“..for a given effect size, the power of the test increased from smallest to the largest sample unit as sample size increased”) is important and could be valuable for readers. Authors should consider adding a figure of these results to the main text for a few select effect sizes (maybe a low and high effect size, or low, medium, and high effect sizes).

Lines 187-188: Please indicate the number or percent of tests that had power values above 0.5 and below 0.4.

Discussion
Overall, the discussion section is very nice.

Line 245: change “costume” to “custom”.

·

Basic reporting

Overall Comments: The authors nicely and succinctly establish both the importance of coral disease and the methods used to study coral disease. Establishing the time of the first reports of coral disease emphasizes the idea that coral diseases at the levels observed is a more recent development, especially when coupled with the rise of other diseases. The establishing of the methods is also critical for the understanding the question posed by the paper. The figures and tables used to support the findings provide a nice visual to ensure that the methods used are both appropriate for the questions asked (figure 1) and provide a representation of the power that ecological methods can have for disease studies.

Use of English: Throughout the provided code and data, there are multiple instances where a mix of English and Spanish are used to annotate code. This should be standardized to English as per PeerJ’s standards and converting all language to English will also better allow readers to understand and interpret the methods that the authors used.

Use of references: The use of references throughout the paper is well done in most cases, except in lines 65-68. Here, the authors explain how coral disease prevalence is obtained using modified ecological techniques without referencing papers that exhibit this method. This can be simply solved by referencing Table. The references should also be checked for completeness, as the reference for Lamb et al 2018 is incomplete and missing 5 authors (Lines 363-364).

Experimental design

Overall comments: The authors employ a simulation strategy that, given the available information, best represents a ‘realistic’ scenario. The use of simulating different distribution patterns and comparing simulated samples to the parameters used to create the simulation allows following results to be credible based on the given parameters. The following tests for accuracy are valid given the types of data simulated and the checks for the appropriateness of the models used is statistically sound. This paper also falls nicely in line with the aim and scope of PeerJ as a biologically motivated paper.

Use of a single coral density and disease prevalence: The authors used a single estimate of coral abundance and disease prevalence from which they based their simulations on. For this, they chose a large, reef-building coral with low abundance. This creates a naturally low sample size, and would thus result in the presented results (Button et al. 2013). The use of this then could be considered cherry picking, as there are other areas of the world such as the Coral Triangle that have a much greater abundance (using cover as a proxy) and diversity of corals (Knowlton & Jackson 2008) and thus would have a greater sample size for a given transect length. This also includes the idea that not all studies survey for a single coral species, and surveying all corals on a smaller transect could lead to larger sample sizes than only surveying a single species on a longer transect. Given the data presented in the paper about Orbicella and Agaricia, the 25x2m transect had on average of 277 corals for just those two species compared to 34 Orbicella. Assuming this average, an area of approximately 400 meters squared (a 200x2 meter transect) would need to be surveyed in one transect to reach the same sample size as studying two species of corals. increasing to a complete survey of all corals present would thus increase the sample size even more. Thus, the authors need to account for variations in abundance to accurately investigate the use of ecological methods to estimate the prevalence of coral disease as in cases of highly abundant corals, smaller transects may be sufficient. (More on this is presented in section 3, validity of the findings).

Pooling of transects: The authors report that a ‘common practice’ of is to pool the counts of diseased and non-diseased corals for the five belt-transects, which leaves each site as a data point. While this practice is employed (eg. Miller 2016), another practice for the analysis of coral disease data from belt-transects is to treat each transect as individual data points (eg. Pollock et al. 2014, Lamb et al. 2018). Here, prevalence is calculated at the transect level and thus provides more data points per site/station with fewer corals per data point. I would suggest that the authors test this alternate method of analysis to better contextualize their study to the literature. Pooling of the transects into a single data point for each 'site' also changes the interpretation of the results as then the unit of study is an area of the reef using transects as a convenient method. With this, 5 10x2 meter transects is equivalent to 5 20x1 meter transects in the amount of area surveyed and thus the results should be similar.

Completeness of methods: In reporting the methods used to estimate disease prevalence, the authors pool the transects in a single station as is reported as ‘best practice’. The reporting of this methodology occurs in the results section (Lines 169-170) and would be better reported in the methods section in the appropriate areas. Another area that should be better represented in the code is the use of the generalized linear models to compare the simulated abundances and prevalence to the original parameters.

Button, K. S., Ioannidis, J. P., Mokrysz, C., Nosek, B. A., Flint, J., Robinson, E. S., & Munafò, M. R. (2013). Power failure: why small sample size undermines the reliability of neuroscience. Nature Reviews Neuroscience, 14(5), 365.

Knowlton, N., & Jackson, J. B. (2008). Shifting baselines, local impacts, and global change on coral reefs. PLoS biology, 6(2), e54.

Lamb, J. B., Willis, B. L., Fiorenza, E. A., Couch, C. S., Howard, R., Rader, D. N., ... & Harvell, C. D. (2018). Plastic waste associated with disease on coral reefs. Science, 359(6374), 460-462.

Miller, M. W., Karazsia, J., Groves, C. E., Griffin, S., Moore, T., Wilber, P., & Gregg, K. (2016). Detecting sedimentation impacts to coral reefs resulting from dredging the Port of Miami, Florida USA. PeerJ, 4, e2711.

Pollock, F. J., Lamb, J. B., Field, S. N., Heron, S. F., Schaffelke, B., Shedrawi, G., ... & Willis, B. L. (2014). Sediment and turbidity associated with offshore dredging increase coral disease prevalence on nearby reefs. PLOS one, 9(7), e102498.

Validity of the findings

Overall comments: The authors bring up an important concern of using methods not designed to adequately sample a population for disease. This allows the authors to make important points about the consequences of finding insignificant differences in a study that could have management concerns. The authors also make a valid recommendation that is to tailor the methods to the questions and species that are being studied. This is an important recommendation as many field studies rely on established sampling protocols (such as transects or quadrats) and may not consider that these methods may not be the most appropriate for the given study. While these conclusions are well stated in the discussion, the authors make claims that overstep the boundaries of their study (detailed below)

Generalization of results: The authors claim that ecological survey methods are insufficient to estimate disease prevalence (Lines 266-267). I would err on the side that, given the data and results presented, this statement is an over-generalization of the results. The authors only simulated data and surveys for one species of coral with one level of disease. In lines 219-220, the authors report that different species of corals have different abundances, which would influence the sample size for a given transect length. To support these claims, I would suggest that the authors vary both abundance and disease prevalence to assess the effects that these variables have on how ecological sampling methods perform in relation to estimating disease prevalence, otherwise the statements made in the discussion and conclusion sections are not fully supported by the results of the simulations presented. Alternatively, the authors can refocus their discussion and conclusions to be focused on low abundance corals. In the current state, the manuscript fails to meet all of the criteria stated for the validity of findings, namely "Conclusion are well stated, linked to original research question & limited to supporting results".

Choice of coral species: As a key reef building coral species that has been recently affected by disease, the choice to simulate data based on Orbicella is understandable. This choice though would limit the scope of the paper to large corals of low abundance. Underlying this data though is naturally lower abundance and cover (Knowlton & Jackson 2008) of corals in the Atlantic Ocean. This choice is biased as the ecological methods would naturally sample fewer corals. I would suggest either that the authors either refocus the discussion to focus on large, low-abundance corals or expand their analyses to include varying densities and abundances of corals as stated above. Also, not all studies focus on single coral species or disease (eg Lamb et al 2018) and could thus circumvent some of the issues underlying using ecological methods to study coral disease. In these instances, all corals in the belt-transect are enumerated and health status determined which could have higher coral abundances than studies focusing on a single species. Surveying for complete coral communities also has the added cost of needing to understand the diversity of coral species at a site

Lamb, J. B., Willis, B. L., Fiorenza, E. A., Couch, C. S., Howard, R., Rader, D. N., ... & Harvell, C. D. (2018). Plastic waste associated with disease on coral reefs. Science, 359(6374), 460-462.

Knowlton, N., & Jackson, J. B. (2008). Shifting baselines, local impacts, and global change on coral reefs. PLoS biology, 6(2), e54.

Additional comments

Overall, the authors present a very important potential shortcoming of coral disease studies in a well organized and clear manner. The methods employed are rigorous for the parameters provided and provide a compelling case for the given study species. The main shortcoming of the study is that the parameters investigated have a small scope while the authors make claims that span beyond the scope of the results they obtained. The authors claim that in all cases, using ecological methods is a poor method for coral disease studies, though they did not examine cases beyond a coral with low abundance and a moderate amount of disease.

---

## Round 0.2 · accepted · Accept

Your paper has been re-evaluated by the most critical referee from the first round, who is now enthusiastic about the publication of your revised paper and recommend acceptance. I find myself in agreement - you have addressed the concerns of both referees from the initial round of review, and I believe that the manuscript is improved by the revisions. I look forward to seeing your manuscript in print, and want to thank you for selecting PeerJ for your paper.

# ·

Basic reporting

Overall Comments: The authors clearly make the point that current methods in disease surveys of key reef building species in the Caribbean may not be sufficient to accurately and reliably measure disease prevalence on a reef. This finding is important as reefs are under a huge amount of stress from anthropogenic sources. Understanding the methods required to reliably and accurately measure disease are then important and this manuscript works on moving the field to a statistically sound foundation.
Use of English: The authors use proficient English throughout the manuscript.
Use of references: The use of references throughout the paper is well done and sufficient to support the claims that they make.

Experimental design

Overall comments: The authors employ a simulation strategy that is used to assess the accuracy and precision of ecological methods in disease surveys on key reef building species. Simulation of data here is important to test these methods and the authors employed a sound step-wise strategy to best capture reality based on the data they had. While this doesn’t address the entire scope of corals and potential distributions, for generally large reef building species, the methods and results are likely to be valid.

Validity of the findings

Overall comments: The authors bring up and address an important concern of using methods not designed to monitor disease. Not only does the authors provide the background in the statistical theory that makes belt transects poor for monitoring disease, they prove the theory using simulation studies. This allows the authors to make important points about how disease surveys of key reef building species are potentially inaccurate which can have implications for management and understanding disease processes. The authors then recommend tailoring study design to the questions being investigated according to study species and general assumptions of the subsequent analyses that will be performed.

Additional comments

Overall, the authors present a very important potential shortcoming of coral disease for large reef building species. The authors adequately investigate the use of ecological methods in disease surveys and find that in some cases, this may not be sufficient.